# OpenReview forum: "AegisLLM: Scaling Agentic Systems for Self-Reflective Defense in LLM Security"
_ICLR.cc/2025/Workshop/BuildingTrust — BuildingTrust_

### Official Review · Reviewer_MEH8 · 2025-02-18
**Modular framework for LLM unlearning and safety, missing ablations & comparisons with baselines**

**Rating:** 5
**Confidence:** 3

**Review:**

This work proposes AegisLLM, a modularized setup for increasing LLM safety and unlearning performance.
It consists of four components: an _orchestrator_ (similar to an input filter), which decides whether a prompt is deemed safe enough to be passed to the _responder_, or whether it should be refused by the _deflector_. If the responder is selected, a fourth component, called _evaluator_ (similar to an output filter), verifies the output of the model, optionally sending it back to the _orchestrator_ for refusal or re-evaluation.
Each component has a specific task and tailored prompt, which is optimized using the DSPy framework.

Pros:
- defense in depth is a reasonable approach for safety
- modular approach enables separation of concerns and separate optimization of each component
- overrefusal-safety trade-off was comprehensively evaluated
- good performance on TOFU

Cons:
- similar frameworks [1-5] were proposed before but are not compared against, making it difficult to assess whether the presented setup is truly effective
- very similar jailbreaking performance as just filtering model outputs with a judge model
- no ablations of the architecture (e.g. what's the impact of excluding the evaluator/orchestrator/deflector,...?), making it difficult to understand which components are significant.
- no information on runtime overhead is given
- unclear how well the components' prompts generalize to other settings (not a huge issue, since they can be dynamically updated)

There are a few minor formatting issues (e.g. in section 4), mostly related to citations, which should be parenthesized when not used as a subject.

[1] Han, Shanshan, et al. "TorchOpera: A Compound AI System for LLM Safety." arXiv preprint arXiv:2406.10847 (2024).

[2] Li, Yuhui, et al. "Rain: Your language models can align themselves without finetuning." arXiv preprint arXiv:2309.07124 (2023).

[3] Wang, Xunguang, et al. "SelfDefend: LLMs Can Defend Themselves against Jailbreaking in a Practical Manner." arXiv preprint arXiv:2406.05498 (2024).

[4] Zeng, Yifan, et al. "Autodefense: Multi-agent llm defense against jailbreak attacks." arXiv preprint arXiv:2403.04783 (2024).

[5] Phute, Mansi, et al. "LLM self defense: By self examination, llms know they are being tricked." arXiv preprint arXiv:2308.07308 (2023).

---

### Official Review · Reviewer_pHCD · 2025-02-26
**A practical agentic security framework for enhancing LLM security and robustness**

**Rating:** 7
**Confidence:** 3

**Review:**

This paper introduces AegisLLM, an agentic security framework designed to enhance LLM robustness against various security threats. This framework is structured as a dynamic, cooperative multi-agent system, comprising autonomous agents such as the orchestrator, deflector, responder, and evaluator, each with specialized functions. By leveraging test-time reasoning and iterative coordination, AegisLLM aims to mitigate risks such as prompt injection, adversarial manipulation, and information leakage. The paper demonstrates the scalability of this approach through the incorporation of additional agent roles and automated prompt optimization using DSPy. The evaluations, particularly on unlearning and jailbreaking benchmarks like WMDP, suggest that AegisLLM outperforms static defenses and exhibits adaptive resilience.

**Clarity:** The paper is generally well-written and structured. The concept of agentic security is clearly introduced, and the roles of the different agents are well-defined. The use of DSPy for prompt optimization is also adequately explained. However, some sections could use more detailed explanations of the specific protocols used for agent communication along with the implementation details of the evaluation metrics.

**Strengths:**
- The agentic security framework is a novel and promising approach to LLM security.
- The framework is shown to be scalable both in terms of adding agent roles and using DSPy for prompt optimization.
- The evaluation results are convincing and properly demonstrate the effectiveness of AegisLLM.

**Weaknesses:**
- The paper could provide more detailed information on the specific protocols used for agent communication.
- More details on the implementation of the evaluation metrics and the experimental setup would be beneficial.
- While the results are promising, more evaluations across a wider range of LLMs and threat scenarios would strengthen the generalizability of the findings.
- The paper does not provide in depth details about the computational overhead associated with the agentic approach.

**Areas of Improvement:**
- Provide more detailed explanations of the agent communication protocols and implementation details.
- Expand the evaluation to include a wider range of LLMs and threat scenarios.
- Include a more detailed analysis of the computational cost associated with AegisLLM.
- Add more analysis of the limitations of the agentic system.

Overall, this paper a novel agentic security framework to enhance LLM security against various threats and is shown to be effective in multiple scenarios. The authors should discuss further on the limitations that this framework and its generalizability to other threat scenarios.

---

### Official Review · Reviewer_eoLt · 2025-03-02
**review of submission 118**

**Rating:** 7
**Confidence:** 4

**Review:**

This paper introduces AegisLLM, a framework that applies agentic systems to LLM security. The idea of conducting security autonomously at inference time with LLM based systems is interesting. The agent architecture also makes sense and is effective. The authors present compelling empirical results across unlearning and jailbreaking benchmarks, demonstrating particular effectiveness on the WMDP benchmark where AegisLLM achieves near-perfect unlearning with minimal examples.

---

### Decision · Program_Chairs · 2025-03-04

Accept